# Bufalin-Mediated Regulation of Cell Signaling Pathways in Different Cancers: Spotlight on JAK/STAT, Wnt/β-Catenin, mTOR, TRAIL/TRAIL-R, and Non-Coding RNAs

**DOI:** 10.3390/molecules28052231

**Published:** 2023-02-27

**Authors:** Ammad Ahmad Farooqi, Venera S. Rakhmetova, Gulnara Kapanova, Gulnara Tashenova, Aigul Tulebayeva, Aida Akhenbekova, Onlassyn Ibekenov, Assiya Turgambayeva, Baojun Xu

**Affiliations:** 1Institute of Biomedical and Genetic Engineering (IBGE), Islamabad 44000, Pakistan; 2Department Internal Diseases, Medical University of Astana, Astana 010000, Kazakhstan; 3Al-Farabi Kazakh National University, 71 Al-Farabi Ave, Almaty 050040, Kazakhstan; 4Scientific Center of Anti-Infectious Drugs, 75 Al-Farabi Ave, Almaty 050040, Kazakhstan; 5Asfendiyarov Kazakh National Medical University, Almaty 050000, Kazakhstan; 6Syzganov National Scientific Center of Surgery, Almaty 050040, Kazakhstan; 7Department of Public Health and Management, NJSC Astana Medical University, Astana 010000, Kazakhstan; 8Food Science and Technology Program, Department of Life Sciences, BNU-HKBU United International College, Zhuhai 519087, China

**Keywords:** cancer, apoptosis, cell signaling, bufalin, metastasis

## Abstract

The renaissance of research into natural products has unequivocally and paradigmatically shifted our knowledge about the significant role of natural products in cancer chemoprevention. Bufalin is a pharmacologically active molecule isolated from the skin of the toad *Bufo gargarizans* or *Bufo melanostictus*. Bufalin has characteristically unique properties to regulate multiple molecular targets and can be used to harness multi-targeted therapeutic regimes against different cancers. There is burgeoning evidence related to functional roles of signaling cascades in carcinogenesis and metastasis. Bufalin has been reported to regulate pleiotropically a myriad of signal transduction cascades in various cancers. Importantly, bufalin mechanistically regulated JAK/STAT, Wnt/β-Catenin, mTOR, TRAIL/TRAIL-R, EGFR, and c-MET pathways. Furthermore, bufalin-mediated modulation of non-coding RNAs in different cancers has also started to gain tremendous momentum. Similarly, bufalin-mediated targeting of tumor microenvironments and tumor macrophages is an area of exciting research and we have only started to scratch the surface of the complicated nature of molecular oncology. Cell culture studies and animal models provide proof-of-concept for the impetus role of bufalin in the inhibition of carcinogenesis and metastasis. Bufalin-related clinical studies are insufficient and interdisciplinary researchers require detailed analysis of the existing knowledge gaps.

## 1. Introduction

Recent advancements in next-generation sequencing and multi-omics analyses have demonstrated how crosstalk of different signaling cascades can result in the formation of a complex web of circuitries within cancer cells that, if fully mapped, can be utilized for more precisely targeted therapies. A wealth of information shows that intracellular signaling is dependent on a multitude of signaling pathways that have evolved for tightly orchestrated and dynamic cellular responses. Furthermore, many signaling cascades interact with each other and form multi-dimensional networks that regulate the integration of numerous inputs to generate sophisticated cellular responses. Seminal research works have uncovered key discoveries in fundamental biology and different types of cellular signaling pathways. Deregulation of transduction cascades not only promoted cancer progression but also fueled the spread of therapeutically resistant and metastatically competent cancer cells to distant organs for the development of secondary tumors [1,2,3,4,5,6]. 

Natural product research in preclinical studies has generated valuable literature related to inhibition of carcinogenesis and metastasis [7,8,9]. Small molecules are pharmacological tools of considerable value for mechanistic dissection of highly intricate biological processes and identification of possible therapeutic interventions. Chemoproteomic workflows have enabled additional multiplexing in research methodologies, which will be valuable for assessing target identification and compound selectivity. The metamorphosis of preclinical research has widened the avenues of effective clinical research. Mechanistic insights gleaned over decades of ground-breaking discoveries have sparked unprecedented research interests in pharmacological evaluation of natural products in the amelioration and remedy of different diseases [10,11]. 

Bufalin is a pharmacologically active molecule isolated from the skin of the toad *Bufo gargarizans* or *Bufo melanostictus*. A substantial volume of conceptual knowledge has been added to the rapidly evolving field of medicinal research associated with the pharmaceutical significance of bufalin. Cancer chemopreventive effects of bufalin have been reviewed previously in various useful and informative review articles [12,13,14,15,16].In the current mini-review we summarize the mechanism-based roles of bufalin in different cancers. We browsed PUBMED and SCOPUS using different keywords to retrieve the results. Moreover, clinical trials associated with bufalin were carefully browsed in https://clinicaltrials.gov access date 15 February 2023. We put the spotlight on bufalin-mediated regulation of JAK/STAT, Wnt/β-Catenin, mTOR, TRAIL/TRAIL-R, EGFR, and c-MET pathways.

## 2. Regulation of JAK/STAT Pathway by Bufalin

A look through a scientific lens indicates that in a burst of research activity, principally published between 1991 and 1994, the cast of JAK-STAT family members and the trajectories of the pathway were mapped to a greater extent. Many functional and structural protein studies and physiological studies on the proteins of the pathway have been reported. The Janus kinase (JAK)/signal transducers and activators of transcription (STATs) transduction pathway is an intracellular signaling cascade required for response to many extracellular ligands. Essentially, phosphorylation induces JAK activation and consequently these kinases phosphorylate intracellular components of the receptors, which allows the recruitment of STAT proteins [17,18,19,20,21]. Genome-wide analyses have yielded a number of discoveries about the biology of STAT proteins. In this section, the most recent evidence has been gathered to summarize multi-step regulation of JAK/STAT pathways by bufalin in different cancers.

Cancer-associated fibroblasts (CAFs) have a critical role in tumor microenvironment. CAF-conditioned media-treated colorectal cancer cells expressed high levels of p-STAT3 and matrix metalloproteinase-2, whereas low levels of E-cadherin were found in hyperactive STAT3-expressing cancer cells. Bufalin blocked CAF-induced invasion and metastasis of colorectal cancer cells by inactivation of the STAT3 pathway. Intraperitoneal injections of bufalin efficiently suppressed hepatic metastatic nodules in mice injected with HCT116 and CAF cells in the spleen (shown in Figure 1) [22].

Bufalin inhibited the growth of the tumors in BALB/c mice subcutaneously injected with CT26 cancer cells. Bufalin notably reduced tumor blood vessels in xenografted mice. There was an evident reduction in the number of blood vessels around subcutaneously transplanted tumors and the proportions of p-STAT3-positive blood vessels in mice intraperitoneally injected with bufalin. Bufalin significantly inhibited liver metastases without affecting bodyweight in rodent models. Small-sized metastatic foci were found in the livers of rodent models treated with bufalin. Importantly, there was significant reduction in metastatic lesions on the surface of the liver in experimental mice administered with intraperitoneal injections of bufalin. Bufalin efficiently reduced the number of blood vessels in the liver and spleen metastases (shown in Figure 1) [23]. 

STAT3 transcriptionally upregulated Bcl2, Mcl-1, survivin, and VEGF (shown in Figure 1). Acetyl-bufalin concentration-dependently reduced p-STAT3 levels in non-small-cell lung cancer cells. CDK9 promoted IL6-induced phosphorylation of STAT3, while IL6-induced STAT3 phosphorylation was considerably impaired in CDK9-silenced cells. Acetyl-bufalin directly interacted with CDK9 and blocked CDK9-mediated activation of STAT3. Intraperitoneal injections of acetyl-bufalin caused regression of primary tumors in rodent models subcutaneously implanted with H460 cancer cells (shown in Figure 1). CDK9-mediated activation of STAT3 was inhibited in the tumor tissues in acetyl-bufalin-treated mice [24].

BF211, a derivative of bufalin, significantly suppressed the levels of p-JAK2 and p-STAT3 in ARP-1 and CAG cells. Tumors were larger in NOD/SCID mice subcutaneously inoculated with ARP-1 multiple myeloma cells. However, BF211 not only reduced tumor growth but also reduced the levels of p-JAK2 and p-STAT3 [25]. 

Bufalin-mediated inactivation of STAT3 has also been reported in colon cancer cells [26]. Bufalin considerably reduced Mcl-1 levels in MCF-7 and MDA-MB-231 cancer cells. Furthermore, bufalin-mediated inactivation of STAT3 remarkably enhanced apoptotic death in breast cancer cells [27]. Overall, these findings indicated that bufalin prominently enhanced apoptosis via inactivation of STAT3. 

Overall, these findings indicate that bufalin demonstrated significant efficiency against STAT3 molecules. Inactivation of STAT3 led to considerable downregulation of the levels of STAT3-regulated target gene networks, which promoted carcinogenesis. There have been encouraging results in the context of significant tumor shrinkage as well as metastatic spread to the secondary organs in mice treated with bufalin. However, there is a need to further explore whether bufalin regulates various other STAT proteins and inhibits carcinogenesis. Therefore, multi-targeted approach by pharmacological targeting of different STAT proteins by bufalin will perhaps be more exciting and valuable in future studies.

## 3. Regulation of AKT/mTOR Pathway by Bufalin

mTOR protein kinase occupies a central role in the nexus of many signaling cascades and plays essential roles in the regulation of different mechanisms. Protein synthesis is a resource-intensive and energy-intensive process in the rapidly growing cells. It is thus tightly controlled by mTORC1, which promotes protein synthesis by phosphorylation of 4E-BPs (eukaryotic initiation factor 4E-binding proteins) and p70S6K1 (S6 kinase 1). In its unphosphorylated state, 4E-BP1 suppressed translation by binding and sequestration of eIF4E (eukaryotic translation initiation factor 4E), an essential constituent of the eIF4F cap-binding complex [28]. mTORC1 regulated cap-dependent translation of mRNAs by direct phosphorylation of the inhibitors of eIF4E: namely, 4E-BP1 and 4E-BP2. TSC2 formed a heterodimeric complex with TSC1 and inhibited mTORC1. However, phosphorylation of TSC2 at the 1462nd threonine by AKT inhibited its GAP activity for RHEB, which therefore remained in a GTP-bound active state and activated mTORC1. AMPK inhibited mTORC1 by phosphorylation of RAPTOR at serine-792 and TSC2 at serine-1387 which promoted the inhibitory functions of TSC1-TSC2 complexes [29,30]. Here, we offer an overview of current advancements in the field regarding the regulation of the AKT/mTOR pathway by bufalin. 

Importantly, bufalin significantly reduced the phosphorylation levels of mTOR and S6K. Furthermore, HIF-1α levels were significantly reduced by bufalin. HIF-1α overexpression attenuated the inhibitory effects of bufalin on ovarian cancer cells. Intraperitoneal injections of bufalin proficiently induced regression of tumor xenografts in rodent models inoculated with PA-1 cells [31]. 

Cbl-b efficiently promoted autophagic pathway activity induced by bufalin through the inactivation of mTOR and activation of ERK1/2. mTOR has been reported to negatively regulate autophagy. Therefore, once activated by AKT/PKB, mTOR inhibited autophagy by enhancing the phosphorylation of p70S6K. Bufalin effectively reduced p-AKT, p-mTOR, and p-p70S6K (Figure 2) [32]. Together, these details indicate that inactivation of AKT/mTOR/p70S6K cascades and functionalization of the ERK pathway are involved in the activation of the autophagic pathway in bufalin-treated MGC803 cancer cells. 

Bufalin inactivated the AKT/mTOR pathway and inhibited migratory and invasive capabilities of ACHN cells. Additionally, bufalin suppressed invasive properties by reducing the levels of HIF-1α and N-cadherin in ACHN cells [33].

There are direct pieces of evidence which suggest that bufalin induces apoptosis via inactivation of AKT/mTOR. Bufalin suppressed the levels of mTOR, p-p70S6K, and p-4EBP1 in Eca109 cancer cells (Figure 2). Bufalin also attenuated growth of orthotopically transplanted tumors in nude mice [34].

Bufalin and sorafenib worked effectively and reduced the levels of p-AKT and p-mTOR in SMMC-7721 cancer cells. Moreover, bufalin and sorafenib remarkably impaired the growth of the volume and mass of primary tumors in mice implanted subcutaneously with SMMC-7721 cells [35]. 

A series of pioneering research works clearly revealed that p70S6K-mediated phosphorylation of S6 correlated with the rate of mRNA translation. Classically, p70S6K has been characterized as a versatile kinase for the regulation of mRNA translation through S6 phosphorylation, and pharmacological targeting of p70S6K has been found to result in effective inhibition of cancer progression. As p70S6K is a downstream effector of mTOR, inactivation of the AKT/mTOR pathway therefore efficiently interferes with the activation of p70S6K, leading to significant reduction in cancer growth in animal models.

## 4. Regulation of Wnt/β-Catenin by Bufalin

In the absence of Wnt signals, degradation of β-catenin is mediated by a destruction complex consisting of adenomatous polyposis coli (APC), Axin and glycogen synthase kinase-3 (GSK-3β) proteins. Following the binding of Wnt to receptors of Frizzled and LRP families on the cell surface, β-catenin efficiently moved into the nucleus and transcriptionally regulated a myriad of gene networks. Phosphorylation of GSK-3β at serine-9 resulted in the inactivation of GSK-3β. Therefore, GSK-3β inactivation led to activation and transportation of β-catenin to the nucleus [36,37,38,39]. In this section, we highlight the recent breakthroughs that have been made in the field of molecular oncology and discuss how regulation of the Wnt/β-catenin pathway by bufalin will influence ongoing basic research and the design of rationale-based clinical trials to improve the treatment options for cancer patients.

Cell cycle-related kinase (CCRK) acted as an oncogenic master modulator for the activation and nuclear translocation of β-catenin, where it formed a complex with transcriptional factor TCF. Notably, the complex binds to promoter regions of EGFR (epidermal growth factor receptor) and CCND1 (cyclin D1). Bufalin efficiently reduced the levels of CCND1, EGFR, and CCRK. It was shown that CCRK overexpression promoted tumorigenesis by activation of β-catenin/TCF signaling. Subcutaneous inoculation of PLC5 cells into the right flanks of athymic nude mice was used for the construction of the xenograft rodent model. Tumor pieces were implanted into the liver lobes of nude mice for the development of orthotopic models. Bufalin not only reduced CCRK but also decreased nuclear levels of β-catenin in the tumor tissues [40].

Bufalin effectively blocked androgen receptor-mediated transcriptional upregulation of CCRK (cell cycle-related kinase) in HepG2.2.15 and PLC5 cells. Levels of phosphorylated androgen receptor were found to be reduced by bufalin. GSK-3β phosphorylation by CCRK caused activation of β-catenin. Bufalin inhibited HBx-mediated intrahepatic tumorigenicity, and reduced the levels of p-ARSer81, CCRK, p-GSK3βSer9, and active β-catenin in tumor tissues [41]. 

Bufalin markedly inhibited the migratory and invasive capacities of hepatocellular carcinoma cells, and efficiently caused reduction in the levels of p-GSK3βSer9 and active β-catenin in BEL-7402 cells [42]. 

Deregulated expression of β-catenin resulted in the instability of the complexes formed with E-cadherin. Dissociation of β-catenin and E-cadherin resulted in a loss of epithelial characteristics and potently promoted increasingly invasive phenotypes. Bufalin interfered with nuclear transportation of β-catenin in colorectal cancer cells [43]. 

The recent advancements in the characterization of aberrantly activated β-catenin target gene programming in cancer cells also provide exceptional prospects for pharmacological targeting of the oncogenic Wnt/β-catenin pathway. As illustrated by the examples given in this section, comprehensive experimental evaluation of the functional effects of bufalin on the Wnt/β-catenin pathway will be advantageous. 

## 5. Regulation of TRAIL Pathway by Bufalin

Excitingly, a maze of information in the rapidly growing field of apoptosis research has unveiled dichotomously branched pathways, consisting of extrinsic and intrinsic apoptotic pathways. Binding of TRAIL to TRAIL-R1 or TRAIL-R2 results in oligomerization of receptors on the cell membrane and initiation of apoptotic cell death. Following ligand–receptor interactions, FAS associated protein with death domain (FADD) is recruited to death domain motifs within the carboxyl terminus of death receptors. Studies have shown that death inducible signaling complex (DISC) is formed at death receptors by assembly of multi-molecular machinery consisting of FADD and pro-caspase-8, and promotes the functionalization of caspase-8 [44,45,46,47,48,49]. During intrinsic apoptosis, loss of subcellular and submitochondrial compartmentalization triggered the exit of cytochrome c, SMAC/DIABLO, and OMI/HTRA. In this section, we review collected key aspects associated with bufalin-mediated regulation of TRAIL-mediated apoptotic cell death. 

Intriguingly, aggregations of lipid rafts as well as redistribution of death receptors (DR4, DR5) in lipid rafts were identified in bufalin-treated MCF-7 and MDA-MB-231 cancer cells. The findings revealed that lipid raft dysfunction caused resistance against TRAIL, whereas bufalin-mediated redistribution of DR4 and DR5 within lipid rafts significantly contributed to TRAIL-mediated apoptotic death in breast cancer cells. Depletion of cholesterol by methyl-β-cyclodextrin has been a widely used approach. Clustering of DR4 and DR5 was reduced markedly in cancer cells pre-treated with methyl-β-cyclodextrin [50]. 

Studies have yielded convincing evidence that Cbl-b negatively regulated the TRAIL-driven pathway. Cbl-b was downregulated by bufalin in MDA-MB-231 and MCF-7 cancer cells. Essentially, bufalin upregulated the levels of DR4 and DR5 by suppression in the levels of Cbl-b. Bufalin and TRAIL-mediated activation of ERK, JNK, and p38 MAPK was found to be significantly enhanced in Cbl-b-silenced cancer cells [51]. 

Bufalin increased the levels of Bax, cytochrome c, Endonuclease G and AIF (apoptosis-inducing factor). Concomitantly, bufalin reduced Bcl-2 in NPC-TW 076 cells. Additionally, bufalin stimulated the expression levels of TRAIL, DR4, DR5, and FADD [52]. 

The TRAIL pathway contains another protein that blocks caspase activation. Importantly, c-FLIP (cellular FLICE inhibitory protein) is an inactive homologue of caspase-8 that contains a DED but lacks a catalytically active site. Bufalin upregulated the expression of DR5 in T24 cancer cells. Moreover, TRAIL and bufalin efficiently reduced the levels of c-FLIP and XIAP in T24 cancer cells (Figure 3) [53].

Bufalin enhanced the levels of FasL, Fas, cytochrome c, and APAF in NCI-H460 cells (Figure 3). Intraperitoneal injections of bufalin efficiently impaired tumor growth in mice inoculated with NCI-H460 cells [54]. 

Bufalin triggered apoptotic death by inducing an increase in mitochondrial release of cytochrome c. Cyclosporin A, a specified inhibitor of mitochondrial permeability transition pore, impaired bufalin-mediated apoptotic death [55].

Bufalin and 5-FU combinatorially reduced the levels of XIAP, Bcl-2, and Mcl-1 and simultaneously enhanced the levels of Bax and Bad in HCT116 cells [56]. Bufalin also promoted mitochondrial release of SMAC/DIABLO (Figure 2) [57]. 

TRAIL-mediated signaling has been extensively studied with rapid advancements that have drawn widespread appreciation. Above all, clinical trials of TRAIL-based therapeutics have increased significantly. Therefore, it is necessary to unveil additional aspects of TRAIL-mediated signaling likely to be targeted by bufalin in different cancers. Comprehensive evaluation of bufalin as a TRAIL sensitizer will be very valuable in cancer prevention. Therefore, expression analysis of death receptors including regulation of death receptor internalization and degradation in bufalin-treated cell lines will yield insightful information. Likewise, TRAIL-based therapeutics can be combined with bufalin for analysis of tumor inhibition in xenografted animal models.

## 6. Regulation of Non-Coding RNAs by Bufalin

More prominently, extraordinary strides have been made in the achievement of a high-resolution view of mechanistic regulation of cell signaling pathways by non-coding RNAs in different cancers. The widespread alteration of non-coding RNAs demonstrated that deregulation of miRNAs [58,59,60], lncRNAs [61,62,63,64], and circular RNAs [65,66,67] contributed to multiple hallmarks of cancer. In this section, we discuss recent findings in the field, where the emerging landscape gives a better overview of the regulation of non-coding RNAs by bufalin in different cancers. 

### 6.1. Tumor Suppressive Role of miRNAs

hsa-miR-3129 directly targeted CD44 and inhibited ovarian cancer progression. Tumor growth was found to be dramatically suppressed in mice inoculated with miR-3129-CAOV-3 cancer cells. Bufalin-mediated cancer inhibitory effects were noted to be impaired in miR-3129-silenced cancer cells [68]. 

SPARC (secreted protein acidic and rich in cysteine) promoted carcinogenesis and metastasis. SPARC has been shown to be directly targeted by miR-203. SPARC levels were found to be decreased significantly in miR-203-transfected U87 and U251 cells. Bufalin stimulated the expression levels of miRNA-203 in glioma cells [69]. 

BAG5 (Bcl-2-associated athanogene 5) was shown to be negatively regulated by miR-127-3p. Overexpression of miR-127-3p efficiently inhibited cancer growth and increased bufalin sensitivity in epithelial ovarian cancer. Tumor growth was found to be significantly reduced in mice inoculated with miR-127-3p-overexpressing OVCAR-3 cells [70]. 

Bufalin promoted miR-148a-mediated targeting of DNMT1 and proficiently inhibited invasive properties of cancer stem cells derived from primary osteosarcoma cells [71]. 

### 6.2. Oncogenic miRNAs

BBC3 (Bcl2 binding component 3) inhibited the progression of osteosarcoma. However, BBC3 is directly targeted by miR-221 in osteosarcoma cells. Bufalin efficiently downregulated miR-221 and promoted apoptotic death in osteosarcoma cells [72]. 

Bufalin-induced apoptotic death was noted to be remarkably enhanced in miR-183-silenced SKOV3 and ES-2 cells. Furthermore, primary tumors derived from miR-183-silenced SKOV3 cancer cells were noted to be markedly reduced in tumor-bearing mice [73].

There is direct evidence that highlights miRNA-mediated targeting of pro-apoptotic genes. miR-298 downregulated BAX and drastically impaired apoptotic death in gastric cancer cells. Bufalin interfered with miR-298-driven targeting of BAX and enhanced apoptotic death [74]. 

Studies have shown that DNA methyltransferases (DNMTs) epigenetically inactivated oncogenic miR-155-5p. Bufalin downregulated DNMT1 and DNMT3a and contemporaneously increased the levels of miR-155-5p. However, miR-155-5p upregulation led to downregulation of FOXO3A. As FOXO3A is involved in the regulation of apoptosis-associated signaling, therefore, miR-155-5p-mediated targeting of FOXO3A impaired bufalin-induced apoptotic death in cancer cells [75]. In accordance with this approach, keeping in view that bufalin stimulated the expression of miR-155-5p and blocked apoptosis, inhibition of miR-155-5p should maximize the chemopreventive effects of bufalin. 

### 6.3. Long Non-Coding RNAs

SRC-1 acted as an oncogene and promoted the stability of XIST RNA in cancer cells. XIST expression was potently downregulated in SRC-1 knockdown-cells, whereas expression of XIST was increased in SRC-1 overexpressing cancer cells. miR-152 not only directly targeted KLF4 but also acted as a competitive endogenous RNA of XIST. Essentially, KLF4 levels were found to be reduced and miR-152 levels were upregulated in SRC-1 knockdown cells, whereas SRC-1 overexpression reduced miR-152 expression and simultaneously stimulated the levels of KLF4. Tumors derived from SRC1-overexpressing LN229 cells were larger. SRC-1 promoted tumorigenesis of glioblastoma, whereas SRC-1 inhibition efficiently impaired intracranial glioblastoma growth in rodent models. Bufalin restricted the proliferation and sphere-forming abilities of SRC-1-overexpressing glioblastoma cells [76].

NORAD, a lncRNA, has been shown to play a central role in carcinogenesis. Importantly, subcutaneous tumor volumes were found to be significantly reduced in mice inoculated with NORAD-silenced-OVCAR-3 cells. Inhibition of NORAD caused notable reduction in bufalin resistance [77]. 

HOTAIR interfered with miR-520b-mediated targeting of FGFR1 and promoted progression of prostate cancer. HOTAIR overexpression led to reversal of the suppressive effects of bufalin on DU145 and PC3 cancer cells [78]. 

### 6.4. Circular RNAs

Tumor suppressor circular RNAs have started to attract widespread attention because of their tremendous potential. Additionally, research into mediated regulation of circRNAs by natural products has opened new horizons in molecular oncology. Circ_0046264 has been demonstrated to suppress the invasive and metastasizing potential of cancer cells. circ_0046264 acted as a sponge for miR-522-3p and inhibited proliferation of cancer cells. Bufalin inhibited tumor growth in experimental mice but, expectedly, knockdown of circ_0046264 led to abrogation of bufalin-mediated cancer inhibitory effects [79]. 

## 7. Tumor Inhibitory Role of Bufalin: Animal Model Studies

Osimertinib is a third-generation standard-of-care therapy for EGFR mutation-positive advanced non–small-cell lung cancers. Osimertinib caused considerable reduction in the levels of Mcl-1 in HCC827 and PC-9 cells. USP9X and Ku70 have been functionally characterized as Mcl-1 deubiquitinases. Studies had shown that deubiquitinases removed polyubiquitin chains from Mcl-1 and enhanced its stability (Figure 4). Levels of Mcl-1 levels were found to be robustly increased in Ku70-overexpressing PC-9 cells. However, silencing of Ku70 led to a notable reduction in the levels of Mcl-1 and enhanced osimertinib-sensitivity in PC-9/OR cells. Moreover, combinatorial treatment with osimertinib and bufalin significantly downregulated the levels of Ku70 and Mcl-1 in tumor tissues of NSCLC xenograft mouse models [80]. 

Bufalin enhanced proteasome activation and degradation of ATP1A1 (Na+/K+-ATPase α1 subunit). There was an evident reduction in the levels of ATP1A1 in glioblastoma tissues in the reported U87 orthotopic glioma animal model. Bufalin demonstrated a unique ability to enter the brain through the blood-brain barrier and potently induced regression of tumors in the central nervous system [81].

The E2 factor (E2F) family of transcriptional factors plays central roles in the transcriptional regulation of oncogenic networks crucial for carcinogenesis and metastasis. Zinc finger protein 91 (ZFP91), an E3 ubiquitin ligase, degraded E2F2. Bufalin-mediated increases in the polyubiquitination levels of E2F2 were substantially decreased in ZFP91-silenced cancer cells in the presence of proteasome inhibitor (MG-132). ZFP91 served as a specified E3 ligase in poly-ubiquitination and proteasomal degradation of E2F2 (Figure 4). Bufalin effectively enhanced complex formation of E2F2-ZFP91. Intraperitoneal administration of bufalin considerably suppressed E2F2 in the tumor tissues in H22 tumor-bearing mice. Bufalin potently enhanced the infiltration rates of CD4+ and CD8+ T cells in tumor tissues, signifying that bufalin augmented anti-tumor responses [82]. 

Bufalin led to significant increase in the ubiquitylation and degradation of Mcl-1. Bufalin dose-dependently activated GSK-3β by suppressing the levels of p-GSK-3β. Bufalin enhanced GSK-3β-mediated reduction in the levels of Mcl-1 [83]. 

AHSA1 (activator of HSP90 ATPase activity-1) is a co-chaperone of HSP90A and activates ATPase activity of HSP90A. Bufalin induced disassembly of AHSA1 and HSP90 and subsequently reduced the levels of PSMD2 and CDK6 in multiple myeloma cells. KU-177, a selective inhibitor of AHSA1, was also found to inhibit the proliferation of MM cells. KU-177 and bortezomib greatly extended the survival of 5TMM3VT mice. AHSA1 levels were higher in bortezomib-resistant ANBL6 cells. However, bufalin interfered with association of AHSA1 and HSP90 and restored bortezomib sensitivity. Remarkably, KU-177 impaired the tumor-forming abilities of bortezomib-resistant ANBL6 cells [84]. 

Increasingly it is being realized that the functional versatilities of NFAT proteins can be analyzed according to their complicated mechanisms of regulation and their capability to integrate calcium signaling with other pathways. One of the roles of Ca2+ is to regulate calcineurin, which consequently dephosphorylates NFAT proteins and promotes their nuclear accumulation. NFAT transcriptionally enhanced the expression of c-Myc (Figure 4). Bufalin reduced the intracellular calcium concentrations. Bufalin efficiently suppressed the levels of NFATC1 and c-Myc in diffuse large B-cell lymphoma cells. Intraperitoneal injections of cyclosporin A and bufalin markedly inhibited tumor growth in NOD/SCID mice subcutaneously injected with SU-DHL-10 cells. Additionally, levels of NFATC1 and c-Myc were found to be reduced in tumor-bearing mice [85]. 

Syndecans belong to the family of transmembrane heparan sulfate proteoglycans expressed on the surfaces of cells. Syndecan-4 contains a transmembrane domain and a cytoplasmic domain. Structurally, syndecan-4 interacts with its interacting partners through its cytoplasmic domain. SDC4 interacts with DDX23 (DEAD-box helicase 23) through its cytosolically located domain. Bufalin promoted the interactions of syndecan-4 with DDX23 for the regulation of genomic instability via induction of double-strand breaks. SDC4 knockdown significantly reduced the levels of p-ERK in HepG2 cells. Bufalin significantly impaired tumor-forming abilities of HepG2 cells in animal models [86]. 

Achaete-scute-like 2 (ASCL2) fuels the migration and invasion of gastric cancer cells. Bufalin downregulated the levels of ASCL2 in AGS cells. Tumors derived from ASCL2-silenced AGS cancer cells were noted to be significantly reduced in experimental mice. Furthermore, bufalin remarkably suppressed the growth of the tumors in mice inoculated with ASCL2-silenced AGS cells [87].

## 8. Regulation of Tumor Microenvironment by Bufalin

Macrophages have functional plasticity and can be polarized into two characteristically distinct phenotypes for modulation of the tumor microenvironment.

Bufalin induced the activation the NF-κB pathway and triggered upregulation of IFNγ and TNFα. However, constitutive overexpression of p50 in bone-marrow-derived macrophage (BMDMs) markedly counteracted the effects of bufalin and concomitantly reduced the expression of M1-associated genes. Importantly, the proportions of CD163+CD206+ M2 macrophages were found to be sharply increased and caused reversal of the pre-dominant effects of bufalin-primed M1 macrophages. In HCC-bearing animal models, overexpression of p50 led to remarkable impairment in the tumor-inhibitory effects of bufalin. Furthermore, the percentage of tumor-promoting M2 macrophages was found to be enhanced in mice bearing p50-overexpressing tumors. The accumulation of bufalin-induced CD4+ and CD8+ T cells in the tissue microenvironment was also reduced in mice inoculated with p50-overexpressing cancer cells [88]. 

Bufalin efficiently inhibited chemo-resistant cell-mediated polarization of M2 macrophages. Macrophage migration inhibitory factor (MIF) inhibited the polarization of macrophages. Bufalin blocked SRC-3-mediated transcriptional upregulation of MIF and abrogated macrophage polarization [89]. 

First discovered in the late 1990s, activating receptor NKG2D participates in the immunosurveillance of cytotoxic lymphocytes primarily through identification of stress-induced ligands MICA/B on the surfaces of cancer cells. Bufalin upregulated membrane bound-MICA (m-MICA) in liver cancer cells and reduced the levels of soluble s-MICA. Bufalin enhanced the expression of NKG2D in NK-92MI cells. Moreover, expression levels of inhibitory receptors (NKG2A, TIGIT and CTLA-4) were also found to be suppressed in NK-92MI cells. Bufalin reduced the levels of ADAM9 and inhibited the shedding of MICA (Figure 4) [90].

## 9. Nanotechnological Approaches for the Delivery of Bufalin

Previously, it has been reported that bufalin exerted cardiotoxic effects at high doses. Therefore, interdisciplinary researchers are working on different strategies to minimize off-target effects and enhance targeted delivery of bufalin to the tumor sites in animal models. 

Tumor-targeted delivery of bufalin-loaded modified albumin-polymer hybrids have been found to be highly effective against hepatocellular carcinoma. Bufalin-loaded nanoparticles demonstrated considerably enhanced release time in circulation, with improved permeability and retention effects. Albumin-coated nanocomplexes attenuated the side effects of bufalin on weight gain in tumor-bearing mice [91]. 

Furthermore, research has shown that dual targeting immunomicelles loaded with bufalin effectively inhibited HCC. There was a substantial accumulation of bufalin-loaded nano-formulations in tumor tissues in mice subcutaneously injected with SMMC-7721 cells [92]. 

Co-delivery of multiple drugs using nanocarriers has been shown to inhibit tumorigenesis. Lenvatinib and bufalin-loaded nanoparticles caused significant shrinkage of the tumor mass in cholangiocarcinoma-bearing rodent models [93]. 

The fluorescence intensities of paclitaxel and bufalin-loaded polymeric micelles were higher in tumors with extended circulation time and lower systemic toxicity [94].

Multifunctional albumin sub-microspheres displayed superior tumor-targeting properties. Bufalin-loaded nano-formulations significantly attenuated cardiac tissue lesions and inflammations in the myocardial interstitial tissues were not noticed [95].

Folic acid-functionalized metal-organic framework nanoparticles have also been found to be effective carriers of payload. These nanoparticles displayed improved water stability and solubility, high cellular uptake, and enhanced activity against breast cancer cells [96]. 

## 10. Clinical Trials

Huachansu is a sterilized hot water extract of dried toad skin. Major chemical components of Huachansu include indole alkaloids and steroidal cardiac glycosides (bufalin). Importantly, dose-limiting toxicities were not observed with the use of eight-times higher doses of Huachansu. Six patients demonstrated prolonged disease stability [97].

Meanwhile, in another clinical trial it was observed that combinatorial treatment with Huachansu and gemcitabine failed to improve the outcome of locally advanced or metastatic pancreatic cancer patients [98]. However, the lack of efficacy reported in these trials does not exclude the possible cancer chemopreventive role of Huachansu in other different solid malignancies.

Importantly, determination of optimal doses, schedules, patient selection, and combinatorial strategies for bufalin and its derivatives requires continued clinical scientific exploration. 

## 11. Future Directions and Existing Knowledge Gaps

High-throughput methods to investigate protein landscapes have quickly transported molecular biologists into a remarkable era of precision oncology. Despite landmark discoveries, numerous important questions remain unanswered. Bufalin-mediated cancer inhibition is gradually gaining attention, and realization of the full-fledged potential of bufalin with reference to broader analysis of regulation of signaling pathways will prove to be advantageous. Detailed research into SHH/GLI pathway regulation is required.. Certain hints have emerged about regulation of SHH/GLI by bufalin in cancer inhibition [99] but these aspects need to be tested comprehensively in animal model studies. TGF/SMAD signaling has also been tested, but needs to be analyzed in a detailed manner in different cancers. Existing evidence reports that bufalin inhibited the TGFβ-induced migratory capacities of A549 cancer cells. Importantly, SMAD2 and SMAD3 are directly activated by TGFβ receptors, and bufalin exerted inhibitory effects on the SMAD2 and SMAD3 in cancer cells. Furthermore, bufalin interfered with TGF/SMAD signaling by suppression of TβRI and TβRII [100]. As can be concluded from the studies presented here, the early phase of research into the cancer chemopreventive role of bufalin has gained momentum, but this is possibly just the beginning of a new era.

## 12. Concluding Remarks

A wealth of knowledge and evidence has surfaced and started to resolve long-standing questions about important molecular targets. Accordingly, regulatory roles of bioactive molecules from natural sources at the molecular level are becoming more comprehensible. The expanding lexicon of pharmacological properties has offered vast opportunities for scientists from different fields to make therapeutically significant discoveries in the future. A major challenge is our incomplete mechanistic knowledge of human biology and the complex processes that take place within the culture of cancer cells and in rodent models after treatment with bufalin.

## Figures and Tables

**Figure 1 molecules-28-02231-f001:**
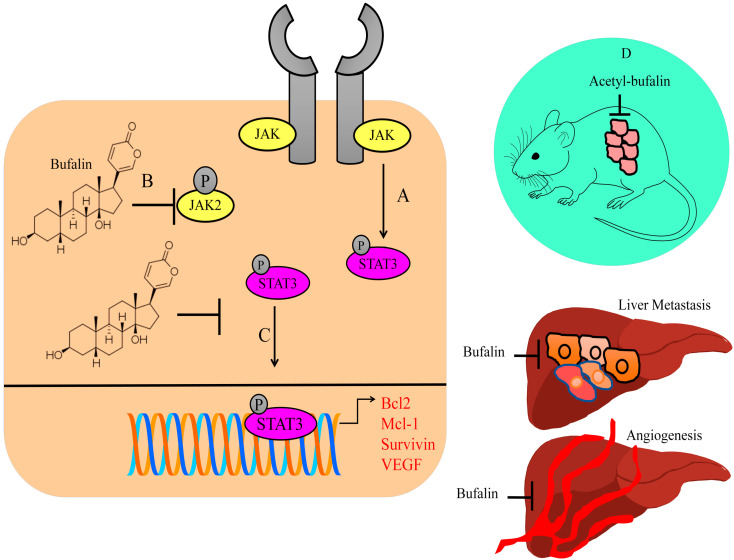
(**A**) JAK/STAT signaling triggered the upregulation of Bcl2, Mcl-1, survivin, and VEGF. JAK2 phosphorylated STAT3 and promoted nuclear accumulation of STAT3 proteins. (**B**,**C**) Bufalin inhibited the activation of JAK2 and STAT3. (**D**) Acetyl-bufalin inhibited tumor formation in mice. Bufalin also inhibited angiogenesis and liver metastasis.

**Figure 2 molecules-28-02231-f002:**
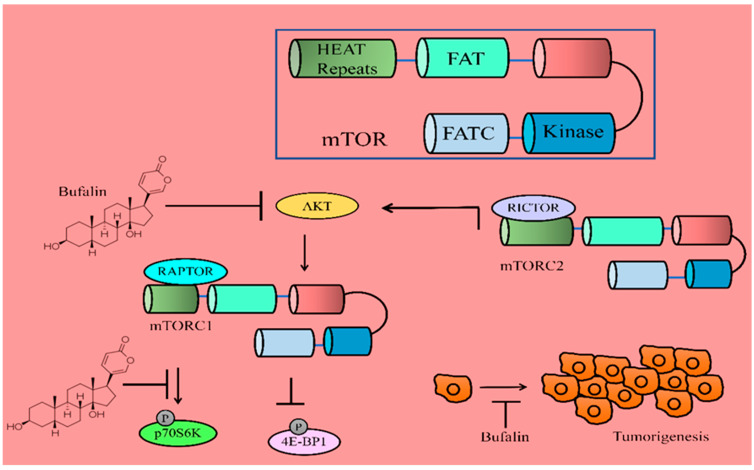
Bufalin mediated inactivation of AKT/mTOR pathway. Bufalin effectively reduced p-AKT, p-mTOR and p-p70S6K. Bufalin inhibited tumor growth by inactivation of AKT/mTOR pathway.

**Figure 3 molecules-28-02231-f003:**
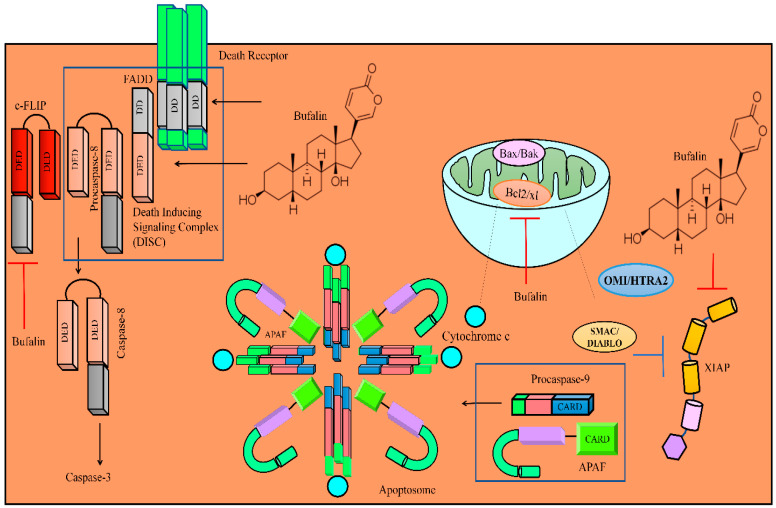
Diagrammatic representation of regulation of TRAIL-mediated apoptotic death by bufalin. Importantly, bufalin enhanced the levels of FADD and DR4/DR5. Bufalin activated death inducible signaling complex. Bufalin triggered the release of cytochrome c and SMAC/DIABLO. C-FLIP prevented the formation of DISC but bufalin prominently reduced the levels of c-FLIP. Bufalin also inhibited Bcl-2 and XIAP.

**Figure 4 molecules-28-02231-f004:**
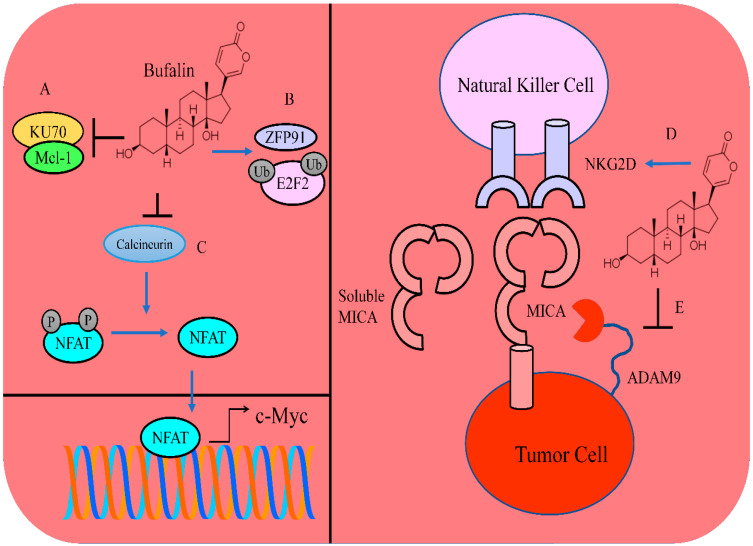
(**A**) Bufalin induced dissociation of KU70 and Mcl-1 and promoted degradation of Mcl-1. (**B**) Bufalin enhanced the interactions of ZFP91 and E2F2. Bufalin mediated an increase in the polyubiquitination levels of E2F2. (**C**) Bufalin inhibited calcineurin mediated dephosphorylation and nuclear accumulation of NFAT. NFAT stimulated the expression of c-Myc. (**D**,**E**) Bufalin enhanced the expression of NKG2D in natural killer cells. Bufalin reduced the levels of ADAM9 and inhibited the shedding of MICA.

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
