# Peer review of "Bufalin-Mediated Regulation of Cell Signaling Pathways in Different Cancers: Spotlight on JAK/STAT, Wnt/β-Catenin, mTOR, TRAIL/TRAIL-R, and Non-Coding RNAs"

_molecules, 2023, doi:10.3390/molecules28052231_

Round 1
Reviewer 1 Report
The authors reviewed Bufalin mediated signal cascade based on several pharmacological effect. I feel those delineations are well-organized and covered all things. Additionally, the authors mentioned future directions and existing knowledge gaps. It is difinitively worth to reading for many readers. I think this perspective can be published after adding the thing below,
In this manuscript, there are no comments of identical activity values. It could be difficult to compare on several activities. I recommend that the authors should add existing activity values on main part.
Author Response
Comment: The authors reviewed Bufalin mediated signal cascade based on several pharmacological effect. I feel those delineations are well-organized and covered all things. Additionally, the authors mentioned future directions and existing knowledge gaps. It is definitively worth to reading for many readers. I think this perspective can be published after adding the thing below,
Response: Thank you for the generous comments. We have extensively edited the manuscript. Added more diagrams and made the text more understandable and easier for the specialist and non-specialist readers.
Comment: In this manuscript, there are no comments of identical activity values. It could be difficult to compare on several activities. I recommend that the authors should add existing activity values on main part.
Response: In animal model studies, majority of findings reported a dosage of 0.1-2 mg/kg intraperitoneally (i.p.) administered at a frequency of once per day 5 days per week to every 3 days for a period ranging from 12 days to 6 weeks in mice xenograft model without inducing adverse effects or significant weight loss.
Reviewer 2 Report
Manuscript describes effect on bufalin on various cell signaling pathways in various cancer cells well. However, figures are not self-explanatory and captions need to described in more detail.
Author Response
Comment: Manuscript describes effect on bufalin on various cell signaling pathways in various cancer cells well. However, figures are not self-explanatory and captions need to described in more detail.
Response: We have added another diagram and added additional diagrams.
Reviewer 3 Report
The current manuscript presented an overview of bufalin-mediated regulation of signaling pathways in cancers. This is interesting, as these signaling pathways could potentially provide targets for targeted therapies for cancers with bufalin. However, the current manuscript should address the following issues before being accepted for publication in the journal.
1) Several review articles have been published regarding the anticancer effects and mechanisms of action. The manuscript needs to discuss the novelty of the current review.
2) The discussion about natural product or natural product research is unnecessary because natural products do not mean less toxicity, better efficacy, or more specificity.
3) Bufalin may regulate the JAK/STAT pathways. However, the exact mechanism of action is not known. This should be discussed in detail.
4) Similarly, how bufalin may regulate the other pathways need to be discussed or clarified.
5) Bufalin has been tested in many previous studies. However, it has not been used for treating cancers. The authors are expected to discuss the issues or potential problems in administrating it to treat cancers or other diseases.
6) The manuscript has many minor language issues that require intensive editing. Here are some examples:
Line17: …understanding about extra-ordinary ability…
Line22: …evidence related to central role of …
Line31: …bufalin but identification of the knowledge…
Line38: … are intertwined to form complex web of circuitries…
Author Response
Comment: The current manuscript presented an overview of bufalin-mediated regulation of signaling pathways in cancers. This is interesting, as these signaling pathways could potentially provide targets for targeted therapies for cancers with bufalin. However, the current manuscript should address the following issues before being accepted for publication in the journal.
Response: Thanks a lot for reviewer’s time input and valuable suggestions. We have done our best to improve the quality of manuscript.
Comment 1) Several review articles have been published regarding the anticancer effects and mechanisms of action. The manuscript needs to discuss the novelty of the current review.
Response: Cancer chemopreventive effects of bufalin have been reviewed previously in some good and informative review articles (12-16). However, in this mini-review we have made efforts to summarize mechanism-based roles of bufalin in different cancers. We have browsed PUBMED and SCOPUS by using different keywords to retrieve the results. Moreover, clinical trials associated with bufalin have also been carefully browsed in https://clinicaltrials.gov/. We have set spotlight on bufalin-mediated regulation of JAK/STAT, Wnt/β-Catenin, mTOR, TRAIL/TRAIL-R, EGFR and c-MET pathways.
Comment 2) The discussion about natural product or natural product research is unnecessary because natural products do not mean less toxicity, better efficacy, or more specificity.
Response: We have extensively edited the draft and added mechanism-based effects of bufalin.
Comment 3) Bufalin may regulate the JAK/STAT pathways. However, the exact mechanism of action is not known. This should be discussed in detail.
Response: We have critically formatted the section and highlighted newly added literature and concluding remarks. We have also added a new diagram.
Comment 4) Similarly, how bufalin may regulate the other pathways need to be discussed or clarified.
Response: We have added further information. We have expanded the regulatory role of bufalin.
Comment 5) Bufalin has been tested in many previous studies. However, it has not been used for treating cancers. The authors are expected to discuss the issues or potential problems in administrating it to treat cancers or other diseases.
Response: We have added the different sections related to CLINICAL TRIALS of Bufalin.
Huachansu is a sterilized hot water extract of dried toad skin. Major chemical components of Huachansu are indole alkaloids and steroidal cardiac glycosides (bufalin). Importantly, dose limiting toxicities were not observed with the use of eight times higher doses of Huachansu. Six patients demonstrated prolonged stable disease (97).
Whereas, in another clinical trial it was observed that combinatorial treatment with Huachansu and gemcitabine failed to improve the outcome of locally advanced or metastatic pancreatic cancer patients (98). However, the lack of efficacies reported in these trials does not exclude the possible cancer chemopreventive role of Huachansu in different other solid malignancies.
Importantly, determination of optimal doses, schedules, patient selection, and combinatorial strategies for bufalin and its derivatives requires continued clinical sci-entific exploration. Large-scale clinical trials are warranted for the translatability of the distilled knowledge for cancer chemopreventive effects of Huachansu and bufalin into clinical applications as safe and effective treatment options for cancer patients in the future.
Comment 6) The manuscript has many minor language issues that require intensive editing. Here are some examples:
Line17: …understanding about extra-ordinary ability…
Renaissance of natural product research has unequivocally and paradigmatically shifted our knowledge about significant role of natural products in cancer chemoprevention.
Line22: …evidence related to central role of …
There is burgeoning evidence related to functional role of signaling cascades in carcinogenesis and metastasis.
Line31: …bufalin but identification of the knowledge…
Although there are scattered pieces of information about cancer chemopreventive role of bufalin, but identification of the gaps in our scientific knowledge
Line38: … are intertwined to form complex web of circuitries…
crosstalk for the formation of a complex web of circuitries
Response: Revision has been done.
Round 2
Reviewer 3 Report
The manuscript has been much improved after the revision. However, the significance of the therapeutic application of bufalin as a "natural product" must be toned down.
Author Response
Comment: The manuscript has been much improved after the revision. However, the significance of the therapeutic application of bufalin as a "natural product" must be toned down.
Response: Part of statements in abract, clinical trials, and conclusion were rewritten.